# A Qualitative and Quantitative Study to Evaluate the Effectiveness and Safety of Magnetic Stimulation in Women with Urinary Incontinence Symptoms and Pelvic Floor Disorders

**DOI:** 10.3390/medicina59050879

**Published:** 2023-05-03

**Authors:** Maurizio Filippini, Nicoletta Biordi, Antonella Curcio, Alessandra Comito, Beatrice Marina Pennati, Miriam Farinelli

**Affiliations:** 1Department of Obstetrics and Gynaecology, Hospital State of Republic of San Marino, 47893 San Marino, San Marino; 2Misericordia Bagno a Ripoli, Gynaecology Unit, 50012 Florence, Italy; 3El. En. Group, Clinical Research & Practice Department, 50041 Calenzano, Italy

**Keywords:** magnetic stimulation, women’s health, urinary incontinence, pelvic floor disorders, ultrasounds, validated questionnaires

## Abstract

*Background and objectives*: Involuntary loss of urine owed to dysfunction of the detrusor muscle or muscles of the pelvic floor is known as urinary incontinence (UI). In this study, ultrasound monitoring was employed for the first time to measure the usefulness and safety of electromagnetic stimulation for women with Stress or Urge UI. *Materials and Methods*: A total of 62 women were enrolled, with a mean age of 55.1 (±14.5); 60% of them were menopausal and presented with urinary incontinence (UI). Eight validated questionnaires were used to evaluate Stress UI, prolapse, overactive bladder urge, faecal incontinence, and quality of life, and the whole study population was tested with ultrasounds at the beginning and at the end of the treatment cycle. The device used was a non-invasive electromagnetic therapeutic system composed of a main unit and an adjustable chair applicator shaped for deep pelvic floor area stimulation. *Results*: Ultrasound measurements and validated questionnaires revealed a consistent and statistically significant (*p* < 0.01) improvement of the mean scores when pre- and post-treatment data were considered. *Conclusions*: Study results showed that the proposed treatment strategy led to a significant improvement in Pelvic Floor Muscle (PFM) tone and strength in patients with UI and pelvic floor disorders, without discomfort or side effects. The demonstration was qualitatively carried out with validated questionnaires and quantitatively with ultrasounds exams. Thus, the “chair” device we used represents valuable and effective support that could be widely employed in the gynaecological field for patients affected by different pathologies.

## 1. Introduction

Loss of urine subject to dysfunction of the detrusor muscle is known as urinary incontinence (UI). It stems from a problem with the pelvic floor and might be related to urethral muscle weakness [1]. Changes that affect the pelvic floor during pregnancy, postpartum and as a result of ageing diminish pelvic support and weaken Pelvic Floor Muscles (PFMs), culminating in UI.

There are many types of UI; however, stress- and urgency-related types are the most common. Stress Urinary Incontinence (SUI) derives from the weakness of urethral and pelvic floor muscles resulting from various factors or traumas, such as damage associated with childbirth, congenital weakness of supporting structures or loss of elasticity with hormone deficiency. In this situation, urine leaks may occur during exercise, coughing, sneezing or lifting weights. Differently, Urge Urinary Incontinence (UUI) derives from involuntary contractions of bladder muscles that start to be overactivated (Overactive Bladder, OAB), clinically resulting in a compelling and sudden need to urinate. When SUI and UUI symptoms coexist, it is known as Mixed Urinary Incontinence (MUI) [2]. UI can seriously influence people’s physical, psychological and social wellbeing and, based on age, it affects 10–20% of all women and up to 70% of elderly women [3]. The prevalence ranges from 17% in women in their 20s to 38% in their 60s [4]; however, only 25% of patients seek specific medical treatment [5]. For this reason, in the last decade, many strategies have been investigated for treating UI [6,7]. They are surgical procedures, physical/conservative therapies (pessaries or vaginal cones, timing urination and restricting fluid intake), pharmacological approaches (anticholinergic drugs, urethral bulking agents, vaginal oestrogens and peripheral nerve stimulation), and behavioural therapies (Kegel exercises, vaginal weight training, biofeedback and electrical stimulation of pelvic floor) [8].

About 90% of patients experience an improvement in incontinence with surgical intervention, through the use of urethral slings or periurethral injections of bulking agents [9]. However, because these treatment options are invasive and may have risks and complications [10], patients are hesitant when considering them. Indeed, some subjects with SUI prefer methods such as electrical stimulation, PFM training, and biofeedback. However, despite the fact that there exist documented advantages of physical therapy, its drawbacks include a slow rate of improvement, low compliance and overall low patient attendance rates [11]. Patients frequently do not practise Kegel exercises correctly or consistently over time, which can reduce their effectiveness (women frequently need to be pushed to conduct Kegel exercises regularly) [12]. Interestingly, it has also been noticed that more than 30% of SUI patients cannot contract their pelvic floor muscles on their first attempt [13].

In this scenario, since 1998, magnetic stimulation, including extracorporeal magnetic stimulation, has been described mainly for UUI, and functional magnetic stimulation for both SUI and MUI have been authorized by the FDA. They are relevant alternatives that have the benefit of allowing people to stay comfortably clothed during a treatment. Recently published studies have proven that TOP Flat Magnetic Stimulation (TOP FMS) technology reduced urge, mixed, and stress incontinence issues, yielding positive effects and enhancing patients’ quality of life (QOL) without risk [6,14]. Indeed, FMS technology, with a homogenous profile and creating no areas of uneven stimulation intensity, enables the strengthening of muscle mass through neuromuscular stimulation because it depolarizes motor neurons, causing large and deep muscle contractions. It is used for muscle rehabilitation as well as for muscular training to build muscle strength.

Due to the minimal impact of the magnetic stimulation on cutaneous receptors, the perceptual pain associated with electrostimulation is also avoided. 

Until recently, magnetic resonance (MRI) was the only imaging method capable of assessing in vivo the anatomic changes and differences of PFMs in normal [15] or traumatic [16,17] conditions. However, MRI is not currently used in clinical settings because of excessive costs and access problems. In contrast, the advent of three-dimensional (3D) ultrasound (US) for the pelvic floor now allows for the evaluation of the patient’s condition with much lower cost for the healthcare system and minimal discomfort to the patient [18,19,20]. Even if the spatial resolution may be poorer, US enables some dynamic multiplanar imaging, which is nearly impossible with MRI technology.

With this study, and for the first time in the scientific scenario, we have empirically evaluated, with the use of US, the effectiveness and safety of FMS to treat women with SUI/UUI symptoms.

## 2. Materials and Methods

### 2.1. Study Population

For this study, 62 women were enrolled, with a mean age of 55.1 (±14.5); 60% of them were menopausal and presented with urinary incontinence (UI). The UI status of the patients was initially evaluated by a gynaecologist, and after an interview addressing symptoms of dysfunction of the pelvic floor and family history of such symptoms and/or surgical procedures, every participant was classified as SUI or UUI based on specific questionnaires according to the UI classification of the International Continence Society [21]. Stress UI was found in 80% of the patients, while 48% showed urgency UI. Moreover, 42.8% presented with pelvic organ prolapse of different grades (I, 53%; II, 47%) (see Table 1).

Eight questionnaires were used to evaluate SUI, prolapse, OAB-urge, faecal incontinence and quality of life at different points in time (see Figure 1). Finally, validated consent and informative papers were submitted to the patients before the treatment began.

Exclusion criteria included, Using the International Continence Society’s Pelvic Organ Prolapse Quantification System (POP-Q) (grade 3), the prolapse of the pelvic organ beyond the hymen; active urinary tract infection with either herpes virus (HSV) or human papillomavirus (HPV); use of diuretics or vaginal oestrogen therapy within the previous 6 months; irregular vaginal bleeding; prior pelvic radiotherapy or surgical procedure for SUI; patients with cardiac pacemakers, implanted defibrillators/neurostimulators, metallic and electronic implants, cardiac disorders, pulmonary insufficiency, malignancy, severe neurological diseases, pregnancy and obesity. Side effects such as temporary muscle spasms, muscular pain, temporary tendon or joint pain, or localized redness or erythema of the skin were assessed before and after the treatment.

### 2.2. Qualitative Evaluation with Validated Questionnaires

Different questionnaires were completed by the patients to evaluate urinary improvement and pelvic floor dysfunction at the beginning, at the fourth session, the sixth session and at the end of the treatment cycle (after 8 sessions). The questionnaires were completed right before every treatment, within the treatment period (until the 8th session). To assess SUI, the Urinary Incontinence Short Form (ICIQ-UI-SF), [6,22] was used to assess the severity of urinary leakage, the clinical manifestations of urinary incontinence, and the impact on quality of life. Moreover, the seven-question Incontinence Impact Questionnaire-Short Form (IIQ-7) [23] was helpful to evaluate the negative impacts of urinary incontinence on health-related quality of life in terms of physical activities, recreation, domestic tasks, social activities, travelling, emotional health and the sensation of frustration [24]. The same questionnaire was used for the OAB condition investigation. In addition, the Incontinence Questionnaire Overactive Bladder Module (ICIQ-OAB) [25] was administered to evaluate overactive bladder and its related impact on quality of life (QoL). Furthermore, the Pelvic Floor Distress Inventory—20 (PFDI-20) questionnaire was considered to assess pelvic organ prolapse distress as well as urinary and anorectal symptoms using three subscales, the Pelvic Organ Prolapse Distress Inventory (POPDI-6), Urinary Distress Inventory (UDI-6) and Colorectal-Anal Distress Inventory (CRADI-8), respectively, [26,27]. Lastly, a questionnaire evaluating the quality of life (I-QoL) [28] of the patients, with 22 questions concerning various aspects of their life, was used. Table 2 shows the details of the questionnaires that were used for every treated patient.

### 2.3. Quantitative Evaluation with Ultrasounds

The whole study population was tested with US by a single clinician before and at the end of the treatment course (8 sessions). A 3D transperineal-translabial ultrasound was carried out while the woman was in a supine position (Figure 2, Figure 3 and Figure 4), using as markers the hyperechoic anterior border of the puborectalis muscle, just posterior to the anorectal muscle and the hyperechogenic posterior surface of the pubic symphysis. A Samsung HERA W9 and WS80 ultrasound (Samsung Healthcare products, South Corea) with a 1–8 MHz 3D volumetric ultrasound transducer (CV1-8A) was used. The core transducer axis was positioned in the mid-sagittal plane, between the two labia majora at the level of the rear fork, with the legs bent at the hips and knees. The maximum transducer 120° angle was chosen as the acquisition angle. Transperineal ultrasound can detect the decrease in anteroposterior (AP) diameter and hiatus area induced by PFM contraction. Within the deep layer, changes in hiatus size and anorectal angle are thought to be caused by relaxation and contraction of the puborectalis muscle. Bladder neck displacement, anorectal angle excursions, levator plate, and hiatus narrowing regarding the inferior border of the pubic symphysis have been widely used in women with pelvic organ prolapse and incontinence as markers of pelvic floor muscle strength. These observations were made right before the image acquisition. It has been discovered that the approach for evaluating pelvic organ descent during the Valsalva manoeuvre well correlates with clinical measurements of descent [29].

### 2.4. Study Device

After the gynaecologist’s clinical picture evaluation, the treatment was performed using the Dr. ARNOLD (DEKA M.e.l.a, Florence, Italy) device. This is a non-invasive therapeutic system CE used since July 2020 and made of a main unit and an adjustable chair applicator shaped for deep pelvic floor area stimulation. The chair is designed for the patient to assume the correct therapeutic posture to maximize the interaction with the electromagnetic stimulation and ensure the best comfort during the treatment session. The patient can keep her clothes on and should sit in the centre of the chair with her spine in an upright position (extension position) and with thighs parallel to the floor, legs perpendicularly flexed and feet flat on the ground (forming an angle of 90° at the knee), avoiding the use of heeled shoes. This way, the perineum of the patient is at the seat centre, which makes it easier for the subject to feel the contraction of the sphincter and pelvic floor muscles during electromagnetic stimulation [30]. The device activity is carried out by selectively stimulating the PFMs with homogeneous-profile (TOP FMS) electromagnetic fields. The recruitment of muscle fibres is made possible by the remarkable uniformity of the magnetic field distribution over a broader area, which prevents the creation of stimulation zones with different intensities. This type of stimulation also provides optimal effects on the blood circulatory system.

### 2.5. Dr. ARNOLD Protocols and Assessments

For this study, two protocols were selected: “Hypotonus/Weakness 1”, where muscles work to improve tone and trophism; and “Hypotonus/Weakness 2”, for stimulating the muscles to increase in volume and strength. Every subject underwent eight sessions, each lasting 30 min and occurring twice a week [31]. According to manufacturer instructions, the first four sessions were performed with an intensity level so that the ideal muscle contraction was reached. Then, if patients showed no discomfort and good tolerance to the treatment, the intensity was raised. At the end of the 8 sessions, all patients were clinically re-evaluated by the study staff, and the questionnaires were re-administered [30].

### 2.6. Statistical Analysis

Student’s *t*-test and SPSS (IBM Corp., New York, NY, USA) were performed to analyse the qualitative and quantitative data obtained. Data were shown as mean ± standard deviation (SD). A *p*-value < 0.01 was considered statistically significant.

## 3. Results

No side effects, such as temporary muscle spasms, muscular pain, temporary tendon or joint pain or localized erythema or reddening of the skin, were reported by the study population before or after the treatment.

### 3.1. Quantitative Evaluation: Ultrasound Measurements

For the quantitative evaluation of the study device effectiveness, all patients in the study were monitored with US before treatment and at the end of the treatment cycle (Figure 5). The distance between the echogenic posterior surface of the inferior border of the pubic symphysis and the echogenic medial-anterior border of the puborectalis muscle of the levator ani was employed to compare pre-and post-treatment improvement. The results showed a statistically significant (*p* < 0.01) distance reduction both at rest (from 59.74 mm ± 7.05 to 56.37 mm ± 8.14) and in a stress (contraction) condition (53.31 mm ± 8.47 to 49.44 mm ± 8.98) (see Table 3).

### 3.2. Qualitative Evaluation: Questionnaire Findings

#### SUI

Questionnaires regarding the SUI revealed a consistent and statistically significant (*p* < 0.01) reduction of the mean scores when the pre- and post-treatment data are compared, suggesting a progressive improvement in the medical condition. Specifically, the ICIQ-UI-SF questionnaire (score range 0–21) reported a mean score of 12.44 (±5.30) at baseline, decreasing to 6.75 (±6.22) right after the last treatment session. Similarly, the UDI-6 survey (score range 0–100) decreased from a baseline mean score of 49.26 (±20.84) to 20.83 (±24.60) after the 8th session. Even with the IIQ-7 (SUI) inquiry, the mean scores went down from baseline 46.91 (±26.57) to 27.88 (±27.69) at the end of the treatment course (see Table 4).

### 3.3. Pelvic Organ Prolapse

An overall improvement was measured when the pelvic organ prolapse was considered pre- versus post-treatment. The cumulative PFDI-SF20 q. has a score range of 0–300, and it showed a baseline mean value of 115.62 (±23.28), lowered to 88.45 (± 12.18) (*p* < 0.01) at the end of the treatment course. Indeed, when subscale results from the CRADI 8 and POPDI-6 questionnaires were analysed, values were more than halved at the end of the study: from 24.55 (±11.03) (pre-treatment) to 13.39 (±8.59) (*p* < 0.05) (post-treatment); and from 25.69 (±13.51) to 11.45 (±13.66) (*p* < 0.01), respectively (see Table 4).

### 3.4. Overactive Bladder Urge/UUI

For Overactive Bladder Urge evaluation, two questionnaires were used at different points in time. First, the ICIQ-OAB (score range 0–16) reported mean values going from 7.2 (±3.73) at T0 to 4.76 (±3.12) at T8. Similarly, a decrease in IIQ-7 (OAB) (score range 0–100) values from 50.16 (± 31.79) at baseline to 38.92 (± 33.89) after the last treatment session was registered (see Table 4).

### 3.5. Quality of Life

The I-QoL questionnaire (score range 0–110) investigated different aspects concerning changes in patients’ life and affecting their everyday routines. The sociality, sexuality and emotional sphere were considered to provide a better understanding and a deeper look into the consequences of having a urinary or pelvic muscle disorder. After the electromagnetic stimulation treatment, patients reported a significant improvement in their quality of life, from a mean score of 72.05 (±21.03) at T0 up to 89.21 (±20.54) (*p* < 0.01) at T8 (see Table 4).

## 4. Discussion

Many non-invasive techniques are available for treating urinary incontinence and pelvic floor disorders. Physiotherapy has been widely used, but it has the disadvantage of having a slow progression and low patient adherence and compliance to the treatment [10]. The same thing can be said for Kegel exercises, whose effectiveness is reduced because they are frequently performed inconsistently or incorrectly over time by the patients. According to the evidence in the literature [32,33,34], TOP FMS is an alternative method that is effective for urinary incontinence in all its forms (SUI, UUI, MUI) [31]. By concentrating electric currents on neuromuscular tissue and depolarizing neurons, this technology causes powerful PFM contractions. The primary somatic and autonomic innervation of the pelvic floor muscles, vaginal wall and rectum, and urinary bladder and urethra originates from the S2–S4 roots of the sacral nerves. Stimulating these roots is an effective way to control the pelvic organs and modulate the pelvic floor [35].

A fundamental aspect that distinguishes Dr. ARNOLD from other devices is the spatial profile of the electromagnetic stimulation. It is homogenously distributed up to the top borders, it covers a wider area, and the lateral profiles are better expressed. This conformation allows for a deep, symmetrical and homogenous distribution of electromagnetic energy, reaching deep neuronal structures inside the pelvis without superficial dispersion. This physical characteristic of the electric current makes it possible to intercept and stimulate bilaterally, in a very selective way, the point of confluence of the sacral branches of the pudendal nerve (S2–S4), the area of maximum response of the pudendal nerve to detrusor inhibition (Figure 6). 

Electrical neuromodulation of the sacral and pudendal nerves has been reported to be effective for the treatment of overactive bladder (OAB) with or without stress urinary incontinence (SUI) and urge incontinence (UUI), [36,37]. Superficial stimulation of the perineal, intravaginal or intrarectal area or direct stimulation of the pudendal nerve is effective for bladder inhibition [38,39,40,41].

Additionally, myofibril growth, as well as the development of new protein filaments and muscle fibres, directly result in muscle hyperplasia and hence in an increase in muscle strength and endurance [42]. Electromagnetic energy, stimulation and deep penetration of the whole pelvic floor area are the basis of this treatment’s effectiveness. Strengthening pelvic floor muscles with the Dr. ARNOLD system has been demonstrated to be effective in the current study. Our findings on the improvement of the patient’s UI symptoms and quality of life align with Isaza et al. [7], Dominguez et al. [30], Lopopolo et al. [6] and Biondo et al. [14].

Indeed, several studies that have used a control group obtained results comparable to ours. This supports our findings, despite the limitations of the study (reported below). Frigerio et al. (2023) [31] demonstrated that urinary-related quality of life scores improved in women who practiced FMS compared to those obtained by women who practiced Pelvic Floor Muscle Training (such as Kegel exercises). Gonzalez Isaza et al. (2022) [7] showed promising improvement in SUI in magnetic stimulation-treated patients compared to the simulated group (sham). In general, these findings underline that magnetic stimulation is a safe and non-invasive alternative for patients who prefer non-surgical treatments. In this study, amelioration in UI, Pelvic Organ Prolapse and OAB symptomatology and quality of life were examined qualitatively by validated questionnaires and quantitatively with US monitoring. Based on the qualitative assessment, there was an improvement in SUI and UUI symptoms after eight sessions of treatment as demonstrated by the reduction of ICIQ-UI-SF, IIQ-7, ICIQ-OAB and UDI-6 questionnaire mean scores (see Table 4). This implied a positive impact on the patient’s quality of life as well. Indeed, increased sexual satisfaction and better control of urination were also reported. This is the first time that Dr. ARNOLD effectiveness has been quantitatively evaluated with echography. From our results, US demonstrated to be a useful tool, and PFM showed significant improvement both at rest and in a stress condition.

In conclusion, the TOP FMS has significant advantages over other pelvic floor treatment strategies. This technology can be used in combination with pharmacological and non-pharmacological modalities [43]. Additionally, it does not require a probe to stimulate the muscles. The regular emission of the gradually supplied energy allows patients to stay dressed and in an ergonomic seat. Subjects who feel that the muscles are relaxing become more self-aware and resume their normal daily activities right away. Additionally, the device’s ability to work with various protocols makes it useful for treating a variety of pathological disorders linked to UI [14].

### Study Limitations and Future Perspectives

Our future goal is to include a control group for comparison purposes. Moreover, it would be interesting to look into the treatment’s long- and very-long-term effects by extending the study period with follow-ups (months or even years). Indeed, the therapeutic efficacy of the treatment hereby presented can only be assessed after a proper follow-up and comparing the obtained results with a control group.

## 5. Conclusions

Study results show that the treatment strategy led, without discomfort or side effects, to a significant improvement in PFM tone and strength in patients with UI and pelvic floor disorders. The demonstration was qualitatively carried out with validated questionnaires and quantitatively with US exams. Thus, the “chair” device we used represents valuable and effective support that could be widely employed in the gynaecologic field for patients with different pathologies.

## Figures and Tables

**Figure 1 medicina-59-00879-f001:**
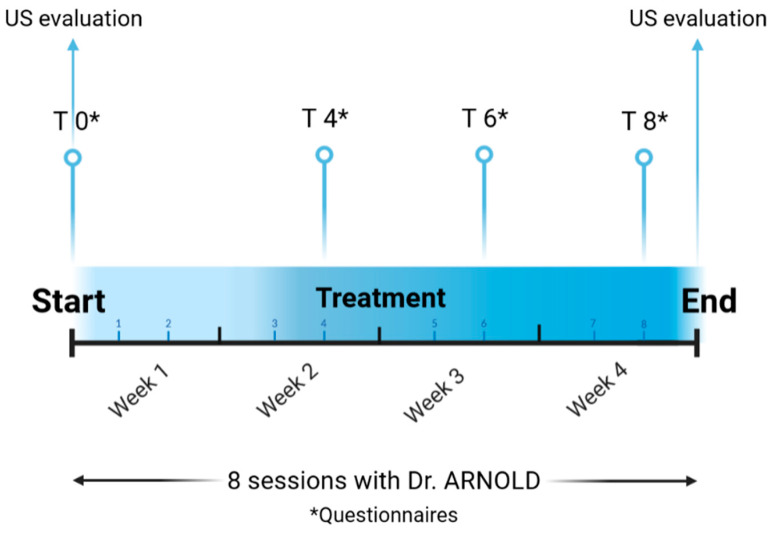
Study Timeline. Every subject was treated for a total of 8 sessions, with two sessions per week. Eight questionnaires (*) were provided at different time points (baseline T0, 4th session T4, 6th session T6, and 8th session T8). The US measurements were performed at baseline and at the end of the treatment course.

**Figure 2 medicina-59-00879-f002:**
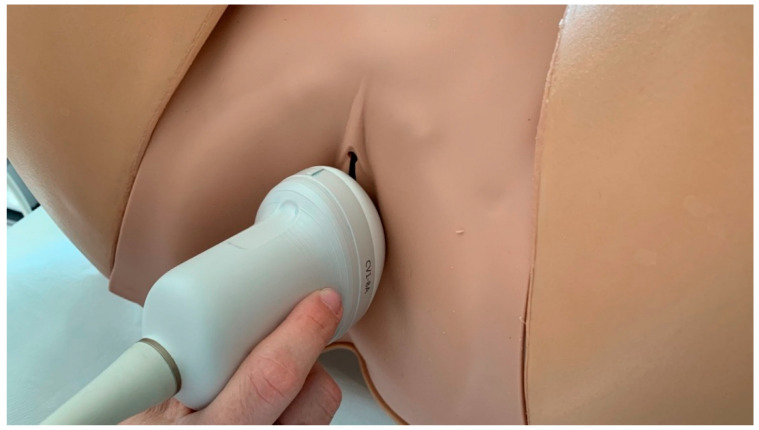
Ultrasound technique for measurements. The patient lays in a dorsal lithotomy position with hips abducted and flexed. A small urine volume in the bladder should be present (100–150 mL) to enable the best view of bladder morphology. No pre-procedure preparation or vaginal/rectal contrast agents are needed. The transducer should be placed with the patient in a neutral posture to prevent undue pressure on surrounding structures and avoid anatomical distortion.

**Figure 3 medicina-59-00879-f003:**
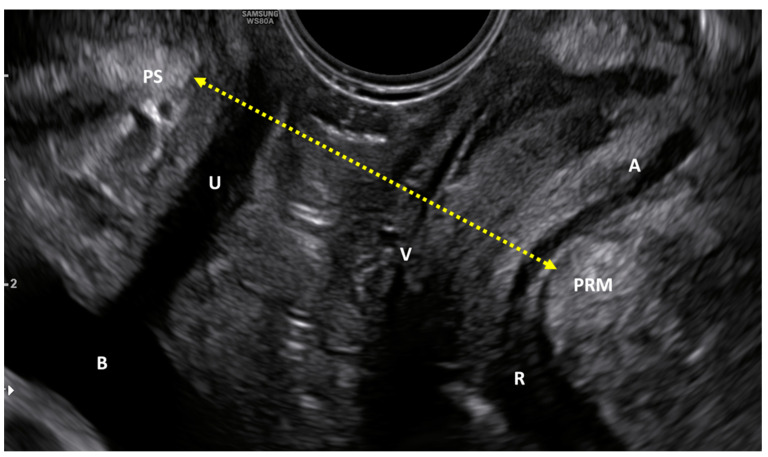
Sample US image of levator ani hiatus. The distance between the inferior border of the pubic symphysis to the medial border of the levator ani (puborectalis muscle) was used to compare pre- and post-treatment improvements. U = Urethra; B = Bladder; PS = Pubic Symphysis; V = Vagina; R = Rectus; A = Anus; PRM = Puborectalis Muscle.

**Figure 4 medicina-59-00879-f004:**
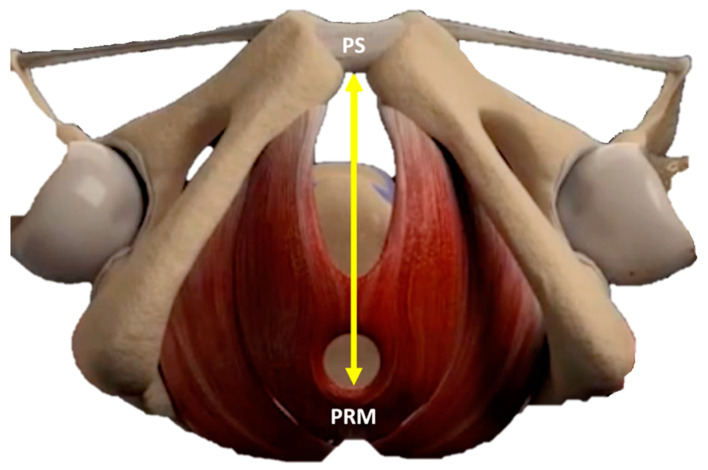
The anatomical distance between the medial edge of the levator ani (puborectalis muscle—PRM) and the lower edge of the pubic symphysis (PS).

**Figure 5 medicina-59-00879-f005:**
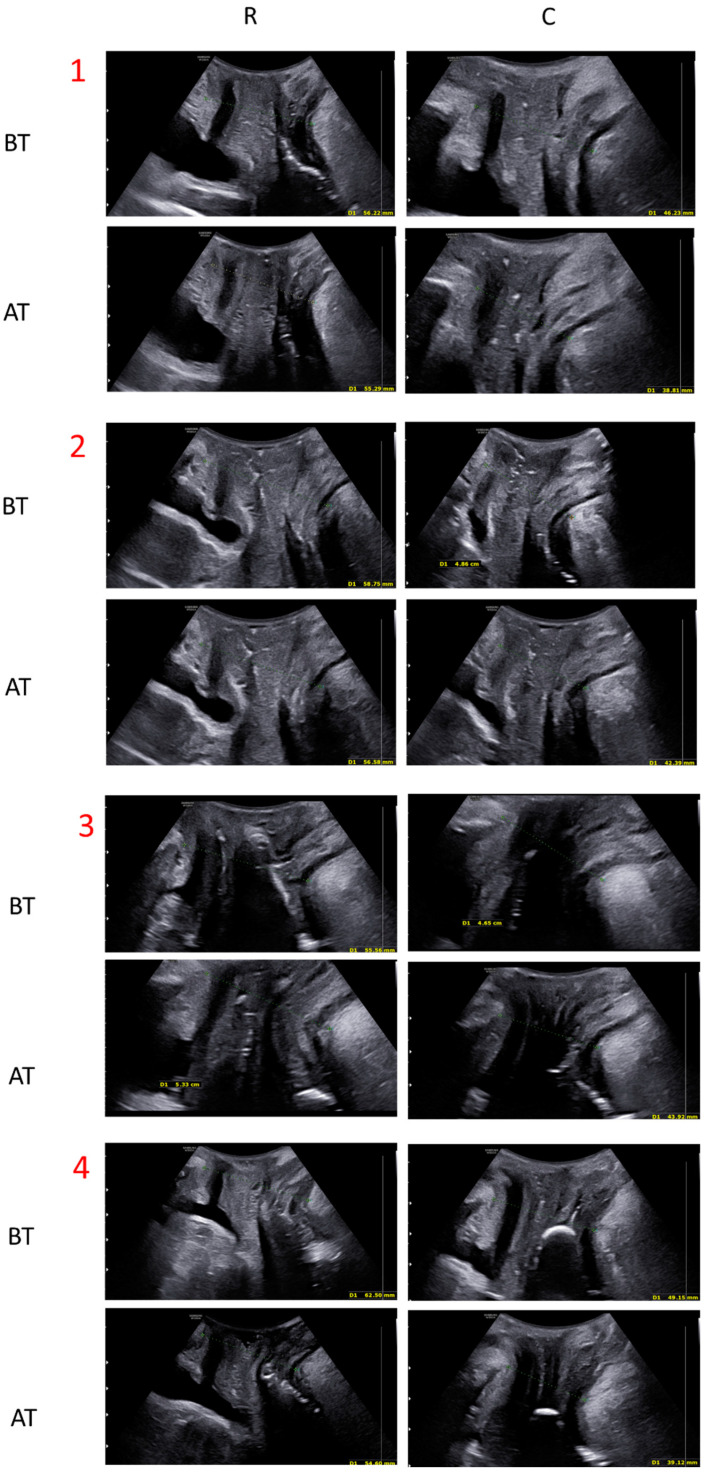
Ultrasound evaluation of four patients before (BT) and after (AT) the treatment with electromagnetic stimulation. For every subject, four pictures were taken. A comparison of a rest (R) condition and under stress/contraction (C) was made.

**Figure 6 medicina-59-00879-f006:**
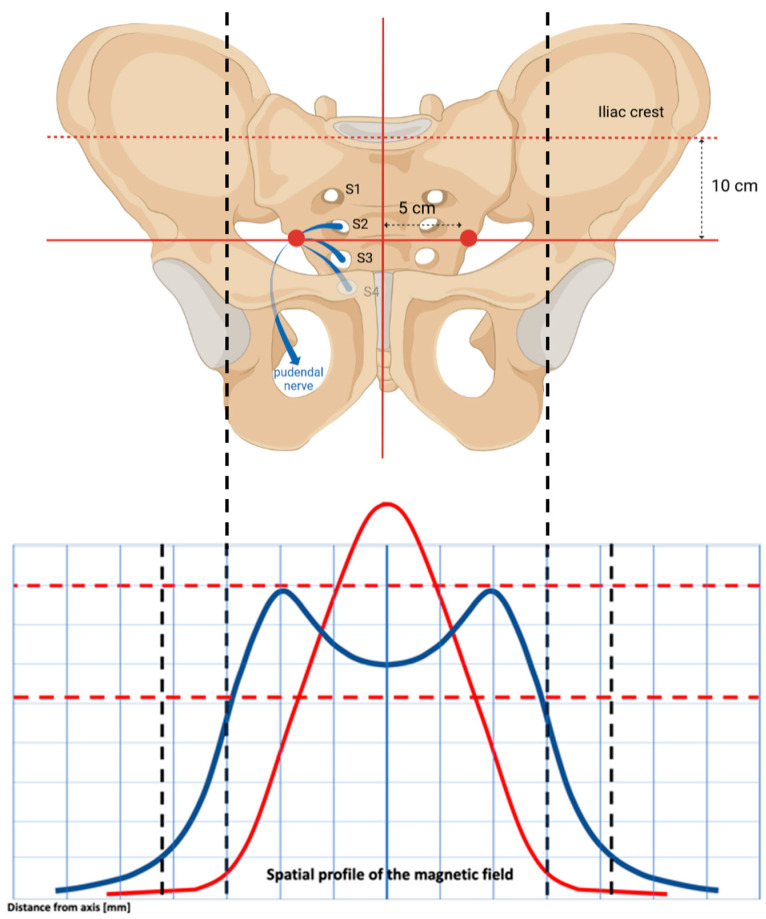
Spatial profile of magnetic field, which allows for a double dome distribution of electromagnetic energy. It intercepts the pudendal nerve at its exit point from the S2–S4 sacral roots. This way, the pudendal nerve is stimulated (as it is shown with red dots, around 10 cm below the iliac crests and 5 cm laterally shifted from the pelvis barycenter) thanks to the homogenous electromagnetic field characterized by a top-flat structure with a diameter of around 16 cm.

**Table 1 medicina-59-00879-t001:** General data describing the study population characteristics. Some patients showed coexisting SUI and UUI symptoms, known as Mixed Urinary Incontinence (MUI).

Number of patients	62	
Average age (mean ± SD)	55.1 ± 14.5	
Menopausal patients (%)	60%	
UI type (%)	SUI	UUI
	80%	48%
Prolapse (%)	42.8%	
	GRADE (% PATIENTS)	
	I (53%) II (47%)	

**Table 2 medicina-59-00879-t002:** Details of the questionnaires used for the study population. They are sorted by medical indication (Pelvic organ Prolapse, Stress Urinary Incontinence, Quality of Life and Overactive Bladder Urge), and the specific name, score range, aim, and administration time is reported for each.

Indication	Questionnaire Name	Score Range	Aim	Administration Time
**SUI**	ICIQ-UI-SF	0–21	Evaluation of clinical manifestations of urinary incontinence, everity of urinary loss, and impact on quality of life	T0, T4, T6, T8
	UDI-6	0–100	Urogenital Distress Inventory in daily life	
	IIQ-7 (SUI)	0–100	Evaluation of the impact of urinary incontinence on activities, relationships and emotional states	
**Pelvic Organ Prolapse**	PFDI-SF20	0–300	Evaluation of the intensity of distress caused by the presence of PFD symptoms	
	POPDI-6	0–100		
	CRADI 8	0–100		
**Overactive Bladder Urge/UUI**	ICIQ-OAB	0–16	For overactive bladder, evaluation of urgency, frequency, nocturia and urgency leakage	
	IIQ-7 (OAB)	0–100	Evaluation of the impact of urinary incontinence on activities, relationships and emotional states	
**Quality of Life**	IQoL	0–110	Evaluation of the impact of urinary/pelvic floor disorder on the everyday quality of life, including socializing, sexuality and emotional states	T0, T8

**Table 3 medicina-59-00879-t003:** Ultrasound quantitative results. Patients were monitored before and at the end of the treatment cycle. The distance (mm) between the echogenic posterior surface of the inferior border of the pubic symphysis and the echogenic medial-anterior border of the puborectalis muscle of the levator ani was employed to compare pre- and post-treatment improvement. General improvement is shown when both the before/after and rest/contraction conditions are compared.

	Rest (mm)	Contraction (mm)
** *Before treatment* **	59.74 ± 7.05	53.31 ± 8.47
** *After treatment* **	56.37 ± 8.14	49.44 ± 8.98

**Table 4 medicina-59-00879-t004:** Questionnaire results cumulative table. Data with standard deviation are reported for every questionnaire at different points in time (baseline and at sessions 4, 6 and 8). ICIQ-UI-SF, IIQ-7 (SUI) and UDI-6 were used for evaluating Stress Urinary Incontinence; PFDI-SF20, POPDI-6 and CRADI 8 for pelvic organ prolapse; ICIQ-OAB and IIQ-7 (OAB) for overactive bladder urge and urge urinary incontinence; and IQoL for the quality-of-life improvement.

	Time Points			
*Questionnaires*	T0	T4	T6	T8
** *ICIQ-UI-SF* **	12.44 ± 5.30	9.57 ± 6.05	7.93 ± 6.11	6.75 ± 6.22
** *IIQ-7 (SUI)* **	46.91 ± 26.57	37.39 ± 28.13	28.80 ± 27.80	27.88 ± 27.69
** *UDI-6* **	49.26 ± 20.84	32.84 ± 19.31	25.73 ± 22.64	20.83 ± 24.60

** *PFDI-SF20* **	115.62 ± 23.28	104.77 ± 23.15	99.56 ± 19.43	88.45 ± 12.18
** *POPDI-6* **	25.69 ± 13.51	18.40 ± 13.58	11.11 ± 10.85	11.45 ± 13.66
** *CRADI 8* **	24.55 ± 11.03	18.75 ± 11.41	15.17 ± 10.58	13.39 ± 8.59

** *ICIQ-OAB* **	7.2 ± 3.73	6.24 ± 2.96	5.48 ± 3.02	4.76 ± 3.12
** *IIQ-7 (OAB)* **	50.16 ± 31.79	45.40 ± 31.72	38.92 ± 35.96	38.92 ± 33.89
** *IQoL* **	72.05 ± 21.03	-	-	89.21 ± 20.54

## Data Availability

Data available on request due to privacy restrictions. The data presented in this study are available on request from the corresponding author.

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
