# Peer review of "A Qualitative and Quantitative Study to Evaluate the Effectiveness and Safety of Magnetic Stimulation in Women with Urinary Incontinence Symptoms and Pelvic Floor Disorders"

_medicina, 2023, doi:10.3390/medicina59050879_

Round 1
Reviewer 1 Report
This paper need some work on the statistical analysis and interpretation of the actual data presented, rather than trying to promote the device.
Why was HSV and HPV excluded?
Was the US measurement automated or was it done by a single clinician? Many clinicians?
It is my understanding from your methods section that the important parameter is the change in displacement, not the change in position. While the distances reduced by 3mm at both rest and stress, this means there is no change in displacement during stress (6.4 vs 6.9mm). What is the relevance of a change in those distances with no change in displacement?
What is the test retest reliability of this method to be able to detect a 3 mm difference? Is this clinically significant according to other studies?
There are lots of p-values quoted on questionnaire results, but no description of what statistical test was performed. There are 9 questionnaires performed at 4 time points but no apparent correction for multiple comparisons. There is also very little discussion/interpretation of these measures.
Table 2 would probably be better as a chart.
The discussion is focused on how the device works without linking it to the results or the interpretation of those results.
Conclusion: you present no convincing evidence that a change in position of 3mm on US is a ‘significant improvement in muscle tone’, especially when displacement is unchanged, nor was there any data presented about safety or comfort to form a conclusion on this.
Author Response
Response to Reviewer 1 Comments
This paper need some work on the statistical analysis and interpretation of the actual data presented, rather than trying to promote the device.
Why was HSV and HPV excluded?
Thank you for the suggestion. In other articles using the same device [i.e. A. Biondo et al (2022), P. Mezzana et al (2022)], patients with an active infection of the urinary tract or with human papillomavirus (HPV) or herpes HSV infections were also excluded. It is a guideline suggested by the device manufacturer.
Was the US measurement automated or was it done by a single clinician? Many clinicians?
Thank you for your comment. We confirm that all US measurements were performed by a single clinician.
It is my understanding from your methods section that the important parameter is the change in displacement, not the change in position. While the distances reduced by 3mm at both rest and stress, this means there is no change in displacement during stress (6.4 vs 6.9mm). What is the relevance of a change in those distances with no change in displacement?
Thank you for your comment. We are not sure that what you ask was between the aims of the study. Indeed, with this study, we wanted to demonstrate that the treatment would have led to a reduction in the distance between the echogenic posterior surface of the inferior border of the pubic symphysis and the echogenic medial-anterior border of the puborectalis muscle of the levator ani when comparing the “rest” situation with the “contracted” one. It is true that before and after the treatment we obtained a very similar reduction of 6.4mm and 6.9mm in the overmentioned distance. But this fact becomes irrelevant if it is considered that even if the difference is similar, the starting values are different (specifically the AT value was smaller ‒ BT 59,74mm vs AT 56,37mm). Indeed, there are many scientific studies such as by Kodikara et al (2019), Verguts et al (2010), and Liang et al (2010) assessing that ultrasound is a highly reliable technique to estimate and calculate the volume of objects and anatomical components. So, it is clear, that even what may appear as a small difference in a linear measurement (3/4mm), becomes way more consistent when the 3D structure is evaluated. Concluding, having a reduction of distances of 3mm at rest and 4mm in contraction, still shows an improvement in the tone and strength of the muscles at issue.
What is the test retest reliability of this method to be able to detect a 3 mm difference? Is this clinically significant according to other studies?
Thank you for the precious suggestion. A dedicated section for the statistical analysis was added to the manuscript. Our study is the first one trying to demonstrate both qualitatively and quantitatively the effect of the device on patients suffering from UI. So, all the other studies using the same instrument (in the “Reference” section they are all mentioned) reported a general clinical improvement in the patients’ conditions regardless of the actual reduction in millimetres.
There are lots of p-values quoted on questionnaire results, but no description of what statistical test was performed. There are 9 questionnaires performed at 4 time points but no apparent correction for multiple comparisons. There is also very little discussion/interpretation of these measures.
Thank you for the precious suggestion. A dedicated section for the statistical analysis was added to the manuscript.
Table 2 would probably be better as a chart.
Thank you for the suggestion. We decided to put a table version because the chart was not as understandable and clear as expected. Anyway, when possible, we will take for sure this suggestion into consideration for future studies.
The discussion is focused on how the device works without linking it to the results or the interpretation of those results.
Conclusion: you present no convincing evidence that a change in position of 3mm on US is a ‘significant improvement in muscle tone’, especially when displacement is unchanged, nor was there any data presented about safety or comfort to form a conclusion on this.

Reviewer 2 Report
13,14,16 (abstract) and 35-36 (introduction) the definition of urinary incontinence reported by authors is wrong:
They stated: "The unintentional leak of urine in the absence of a detrusor muscle contraction within the bladder is known as urinary incontinence": this is the definition of stress urinary incontinence, a specific type on incontinence and not of urinary incontinence. According to International Continence Society Terminology Standard Report [2022], the urinary incontinence is described as "any involuntary loss of urine" . (need correction).
The results are interesting but the study limitations, also reported by the Auhors (control group ? too short follow-up) ) reduce the strenght of the conclusion themselves, especially regarding the long-term effectiveness of the procedure (that is more expensive than pelvic floor muscle traning).
The study show the effect of magnetic stimulation on the the pelvic muscle floor (reduction of the distance between the medial edge of levator ani and the lower edge of the pubic symphysis) and the safety of procedure but the terapeutic effecacy could only be assessed after proper follow-up and comparing the results with a control group.
Author Response
Response to Reviewer 2 Comments
13,14,16 (abstract) and 35-36 (introduction) the definition of urinary incontinence reported by authors is wrong:
They stated: "The unintentional leak of urine in the absence of a detrusor muscle contraction within the bladder is known as urinary incontinence": this is the definition of stress urinary incontinence, a specific type on incontinence and not of urinary incontinence. According to International Continence Society Terminology Standard Report [2022], the urinary incontinence is described as "any involuntary loss of urine". (need correction).
Thank you for the suggestion. It has been corrected.
The results are interesting but the study limitations, also reported by the Auhors (control group ? too short follow-up) ) reduce the strenght of the conclusion themselves, especially regarding the long-term effectiveness of the procedure (that is more expensive than pelvic floor muscle traning).
Thank you for the suggestion. We fully agree with what you say. Unfortunately, we did not have the possibility to have a control group and/or long-term follow-ups so it was necessary to mention them as study limitations even if this would have possibly weakened the study results.
The study show the effect of magnetic stimulation on the the pelvic muscle floor (reduction of the distance between the medial edge of levator ani and the lower edge of the pubic symphysis) and the safety of procedure but the terapeutic effecacy could only be assessed after proper follow-up and comparing the results with a control group.
Thank you for the suggestion. We fully agree with your statement. In the “Study limitations” section it was added a brief sentence to emphasize the matter.

Reviewer 3 Report
I think the study is overall well conducted and I really appreciated the fact that, other than PRO questionnaires US measurements were evaluated.
I would have probably further discussed the US methodology and explained with some pictures the operative method of measuring and provide examples of pathological/non pathological evaluation in different clinical settings (pop grade I/II/III, UI, SUI, MUI etc...)
PFM line 26: please explain the acronym before using the abbreviation
it induces PFM malfunction and directly causes UI; line 40, I would suggest rephrasing into “ when enough support is not guaranteed(...)PFM malfunction occurs and causes UI”, in order to link the subject to the main verb, avoiding the repetition of “it” pronoun every, line 103, capital letter after full stop mark
lines 97-109 I would consider adding a descriptive table of the baseline conditions of the study population, considering also which percentages of the population reported which symptoms and explaining percentages reported when needed e.g. POP grades
“An adequate urine volume in the bladder should be present” line 173, please explain how much urinary volume should be present and maybe summarize if and how it could change the ultrasound evaluation
“reduction both at rest (from 223 59,74 mm ± 7,05 to 56,37mm ± 8,14) and in stress (contraction) condition (53,31mm ± 8,47 224 to 49,44mm ± 8,98)” lines 223-225 I would add a simple table to summarize the US findings and I would specify if US, being an operator dependent imaging, was always performed by the same operator and if not how many operators were involved.
I would also add an example of measurements shown as a figure in the manuscript, in order to provide a clear and effective example of how US measurements are take and how they differ BT and AT
Line 254 add the reference to the table 2
Line 274 “TOP FMS is an alternative way 273 to be effective on urinary incontinence in all its forms”: could you please provide some explanation about the reason why POP grade III was mentioned in the exclusion criteria, given the piece of information provided at line 273-274?
Line 299 “TOP HAT structure “could you please explain the acronym
Minor English revision to be provided.
Probably would have also further discussed benefit of this new device compared to existing ones

Author Response
Response to Reviewer 3 Comments
I think the study is overall well conducted and I really appreciated the fact that, other than PRO questionnaires US measurements were evaluated.
I would have probably further discussed the US methodology and explained with some pictures the operative method of measuring and provide examples of pathological/non pathological evaluation in different clinical settings (pop grade I/II/III, UI, SUI, MUI etc...)
Thank you for your comment. For sure we will take this suggestion in consideration for future studies.
PFM line 26: please explain the acronym before using the abbreviation
Thank you for the suggestion. It has been corrected.
It induces PFM malfunction and directly causes UI; line 40, I would suggest rephrasing into “ when enough support is not guaranteed(...)PFM malfunction occurs and causes UI”, in order to link the subject to the main verb, avoiding the repetition of “it” pronoun every, line 103, capital letter after full stop mark
Thank you for the suggestion. It has been corrected.
lines 97-109 I would consider adding a descriptive table of the baseline conditions of the study population, considering also which percentages of the population reported which symptoms and explaining percentages reported when needed e.g. POP grades
Thank you for the suggestion. It has been added to the manuscript (now Table 1).
“An adequate urine volume in the bladder should be present” line 173, please explain how much urinary volume should be present and maybe summarize if and how it could change the ultrasound evaluation
Thank you for the suggestion. It has been added to the manuscript.
“reduction both at rest (from 223 59,74 mm ± 7,05 to 56,37mm ± 8,14) and in stress (contraction) condition (53,31mm ± 8,47 224 to 49,44mm ± 8,98)” lines 223-225 I would add a simple table to summarize the US findings and I would specify if US, being an operator dependent imaging, was always performed by the same operator and if not how many operators were involved.
Thank you for the suggestion. A table was added (now table 3) to summarize the US findings.
I would also add an example of measurements shown as a figure in the manuscript, in order to provide a clear and effective example of how US measurements are take and how they differ BT and AT
Thank you for the suggestion. We already have a figure that can fulfil the characteristics you ask (see Figure 5). Further enlarging the picture, specifically in panels 3 and 4, are visible the measurements of the distance with a green fine line and a value, before and after the treatment.
Line 254 add the reference to the table 2
Thank you for the suggestion. It has been added to the manuscript.
Line 274 “TOP FMS is an alternative way 273 to be effective on urinary incontinence in all its forms”: could you please provide some explanation about the reason why POP grade III was mentioned in the exclusion criteria, given the piece of information provided at line 273-274?
Thank you for the suggestion. With that sentence, we referred to all the forms of incontinence (SUI, UUI, MUI) and not the conditions leading to the problem.
Line 299 “TOP HAT structure “could you please explain the acronym
Thank you for the suggestion. It was a misspelling mistake (It was “top-flat”). It is now been corrected.
Minor English revision to be provided.
Thank you for the suggestion.
Probably would have also further discussed benefit of this new device compared to existing ones

Reviewer 4 Report
Please find my comments to the MS entitled " A qualitative and quantitative study to evaluate the effectiveness and safety of magnetic stimulation in women with urinary incontinence symptoms and pelvic floor disorders" by Filippini et al.
The authors examined whether TOP-flat magnetic stimulation, that elicits muscle contraction, can be used to upgrade the tone and strength of pelvic musculature and thereby to ameliorate urinary incontinence and pelvic floor symptoms. Sixty-two pre- and post-menopausal women with urinary incontinence and pelvic floor dysfunction were subjected to TOP-flat magnetic stimulation treatment at eight time points during a four week treatment interval. Women were asked to complete eight different questionnaires at each time point and ultrasonographic monitoring before and at the end of the treatment were employed. The eight questionnaires used are suitable to evaluate urinary incontinence, prolapse, over-active bladder urge, fecal incontinence, and quality of life. The distance between the echogenic posterior surface of the inferior border of the pubic symphysis and the echogenic medial-anterior border of the puborectalis muscle of the levator ani was compared pre-and post-treatment. Reduction in distances were statistically significant. Questionnaires provided positive results.
Main comment: The English language as regards grammar is incoherent. Examples are provided but every second line needs to be rephrased. Here are some examples.
Title would read better as such: A qualitative and quantitative study to evaluate the effectiveness and safety of magnetic stimulation in women with urinary incontinence and pelvic floor dysfunction".
Abstract: line 1, LEAK is unintentional – start sentence with " The leak …”. Line 3 – delete scientific scenario
Line 25: the treatment strategy we proposed led without discomfort or side effects to a significant improvement – please rephrase for clarity.
Line 25: "different pathologies is too wide – please specify"
Introduction: line 35, LEAK is unintentional by definition please rephrase.
Line 36: replace the word "absence" by dysfunction or weakening.
Line 37: omit for example.
Line 43: consequent not consequently.
Line 44: replace with by due to.
Line 46: omit differently – not needed.
Line 61: "through the use" rephrase
Line 64: "But, despite there are documented advantages" please rephrase.
Line 74: omit "a".
Line 76: "recently published amply proven that"..rephrase coherently.
line 93 – delete scientific scenario.
The methodology is sound and the discussion acceptable.
A comparison between pre- and post-menopausal women should be considered.
However, the MS needs to be re-submitted in proper language and needs to be rephrased and re-edited.
Author Response
Response to Reviewer 4 Comments
Please find my comments to the MS entitled " A qualitative and quantitative study to evaluate the effectiveness and safety of magnetic stimulation in women with urinary incontinence symptoms and pelvic floor disorders" by Filippini et al.
The authors examined whether TOP-flat magnetic stimulation, that elicits muscle contraction, can be used to upgrade the tone and strength of pelvic musculature and thereby to ameliorate urinary incontinence and pelvic floor symptoms. Sixty-two pre- and post-menopausal women with urinary incontinence and pelvic floor dysfunction were subjected to TOP-flat magnetic stimulation treatment at eight time points during a four week treatment interval. Women were asked to complete eight different questionnaires at each time point and ultrasonographic monitoring before and at the end of the treatment were employed. The eight questionnaires used are suitable to evaluate urinary incontinence, prolapse, over-active bladder urge, fecal incontinence, and quality of life. The distance between the echogenic posterior surface of the inferior border of the pubic symphysis and the echogenic medial-anterior border of the puborectalis muscle of the levator ani was compared pre-and post-treatment. Reduction in distances were statistically significant. Questionnaires provided positive results.
Main comment: The English language as regards grammar is incoherent. Examples are provided but every second line needs to be rephrased. Here are some examples.
Title would read better as such: A qualitative and quantitative study to evaluate the effectiveness and safety of magnetic stimulation in women with urinary incontinence and pelvic floor dysfunction".
Abstract: line 1, LEAK is unintentional – start sentence with " The leak …”. Line 3 – delete scientific scenario
Line 25: the treatment strategy we proposed led without discomfort or side effects to a significant improvement – please rephrase for clarity.
Line 25: "different pathologies is too wide – please specify"
Introduction: line 35, LEAK is unintentional by definition please rephrase.
Line 36: replace the word "absence" by dysfunction or weakening.
Line 37: omit for example.
Line 43: consequent not consequently.
Line 44: replace with by due to.
Line 46: omit differently – not needed.
Line 61: "through the use" rephrase
Line 64: "But, despite there are documented advantages" please rephrase.
Line 74: omit "a".
Line 76: "recently published amply proven that"..rephrase coherently.
line 93 – delete scientific scenario.
Thank you for all the comments. We had the manuscript checked for proper English use.
The methodology is sound and the discussion acceptable.
A comparison between pre- and post-menopausal women should be considered.
Thank you for your suggestion. We are considering this aspect for a future study.
However, the MS needs to be re-submitted in proper language and needs to be rephrased and re-edited.

Round 2
Reviewer 1 Report
Please find my comments to the revision provided to the MS entitled " A qualitative and quantitative study to evaluate the effectiveness and safety of magnetic stimulation in women with urinary incontinence symptoms and pelvic floor disorders".
The authors examined whether TOP-flat magnetic stimulation, that elicits muscle contraction, can be used to upgrade the tone and strength of pelvic musculature and thereby to ameliorate urinary incontinence and pelvic floor symptoms. The methodology and results are however sound. Please revise according to the following comments.
Abstract: The first sentence in the abstract reads: "Any involuntary loss of urine in the absence of a detrusor muscle contraction within the bladder is known as urinary incontinence (UI)" please rephrase as such: Involuntary loss of urine owed to dysfunction of the detrusor muscle or muscles of the pelvic floor is known as urinary incontinence (UI)"
2cd sentence:
With this study, and for the first time in the scientific scenario, we have empirically evaluated the effectiveness and safety of electromagnetic stimulation to treat women with Stress or Urge UI symptoms with ultrasounds rephrase as such In this study ultrasound monitoring was employed for the first time to measure the usefulness and safety of electromagnetic stimulation for the of women with Stress or Urge UI.
Introduction:
The first sentence in the abstract re: "Any loss of urine of urine in the absence of a detrusor muscle contraction within the bladder is known as urinary incontinence (UI)" rephrase as such: loss of urine subject to dysfunction of the detrusor muscle is known as urinary incontinence (UI)"
2cd sentence: "For example, complex changes affect the pelvic floor during pregnancy, postpartum, and ageing, and when enough support for pelvic organs and continence mechanisms is not guaranteed by the Pelvic Floor Muscles (PFMs), PFM malfunction occurs and directly causes UI". rephrase as such: " Changes that affect the pelvic floor during pregnancy, postpartum and at ageing diminish pelvic support and weaken Pelvic Floor Muscles (PFMs) culminate in UI".
Discussion: Please omit the term "for example" wherever introduced. Not needed.
Page 13: new paragraph introduced by authors reads: "Indeed, there are several scientific studies that have used a control group and obtained results comparable to ours. This supports our findings despite the limitations of the study (reported below). For example, Frigerio et al. (2023) [31] observed improvements in urinary-related quality of life scores after FMS compared to those of women who underwent Pelvic Floor Muscle Training (such as Kegel exercises). Also, Gonzalez Isaza et al (2022) [7] reported promising improvement in SUI in magnetic stimulation-treated patients compared to the simulated group (sham). In general, these findings underline that magnetic stimulation is a safe and non-invasive alternative for patients who prefer non-surgical treatments.
rephrase as such: "Indeed, several studies that have used a control group obtained results comparable to ours. This supports our findings despite the limitations of the study (reported below). Frigerio et al. (2023) [31] demonstrated that urinary-related quality of life scores improved in women who practiced FMS compared to those obtained by women who practiced Pelvic Floor Muscle Training (such as Kegel exercises). Gonzalez Isaza et al. (2022) [7] showed promising improvement in SUI in magnetic stimulation-treated patients compared to the simulated group (sham). In general, these findings underline that magnetic stimulation is a safe and non-invasive alternative for patients who prefer non-surgical treatments.
2nd paragraph reads: Therefore, in this study, in UI, Pelvic Organ Prolapse, OAB and quality of life conditions were determined qualitatively by validated questionnaires and quantitatively with US, omitoring.
rephrase as such: In this study, amelioration in UI, Pelvic Organ Prolapse and OAB symptomatology and quality of life were examined qualitatively by validated questionnaires and quantitatively with US monitoring.
Last paragraph reads: Firstly, this technology can be also used in combination with pharmacological and non-pharmacological approaches [43]. Moreover, it does not require the use of a probe to stimulate the muscles, and due to the regular emission of the gradually supplied energy, it enables patients to stay completely clothed in an ergonomic seat. Subjects are able to feel the muscles relaxing during therapy, which helps them become more self aware and resume their normal daily activities right away. Additionally, the device's ability to work with various protocols makes it useful for treating a variety of pathological disorders associated with UI [14].
rephrase as such: This technology can be used in combination with pharmacological and non-pharmacological modalities [43]. Additionally, it does not require a probe to stimulate the muscles. The regular emission of the gradually supplied energy allow patients to stay dressed in an ergonomic seat. Subjects who feel that the muscles are relaxing become more self- aware and resume their normal daily activities right away. Additionally, the device's ability to work with various protocols makes it useful for treating a variety of pathological disorders linked to UI [14].
Other changes introduced are acceptable
